# Sustaining Investigative Audit Quality through Auditor Competency and Digital Forensic Support: A Consensus Study

**Hendra Susanto [1],\*, Sri Mulyani [1], Citra Sukmadilaga [2] and Erlane K. Ghani [3],\***

1   Faculty of Economics and Business, University of Singaperbangsa Karawang, Karawang 41361, Indonesia
2   Faculty of Economics and Business, Padjadjaran University, Bandung 45363, Indonesia
3   Faculty of Accountancy, Universiti Teknologi MARA, Puncak Alam 42300, Malaysia
\*   Correspondence: hendra.susanto@bpk.go.id (H.S.); erlanekg@uitm.edu.my (E.K.G.)

**Abstract:** The increased public awareness of the impact of fraudulent activities has put pressure on corporations to practise better corporate behaviour. As a result, their stakeholders demanded that corporations increase the level of transparency that pertains to their corporate behaviour and provide them with sustainable assurance. One of the ways that they can improve the way they conduct business is by ensuring that their investigative audits are of a high quality. In this study, we investigate the factors that influence the quality of investigative audits. In particular, two factors are chosen, namely, auditor competency and digital forensic support. Using a questionnaire survey as the research instrument, the questionnaires were distributed to 150 investigative auditors who worked for the Indonesian Audit Investigative Board (BPK). This study shows that both factors significantly and positively influence investigative audit quality. The findings of this study can help related parties to better understanding the factors that contribute to investigative auditing and, as a consequence, suggest ways to improve the investigative audit quality. For BPK, which has the authority to conduct audits of the management and accountability of state finances, the findings serve as a fundamental insight into sustaining work integrity and professionalism.

**Keywords:** auditor competency; digital forensic support; investigative audit; Indonesia

## 1. Introduction

In Indonesia, the Audit Board of the Republic of Indonesia (BPK) is given the mandate to audit state finances by Law No.15 of 2004, including the authority to perform financial audits, performance audits, and audits with specific objectives such as investigative audits. Investigative audits aim to reveal indications of state/regional losses and/or crime. When indications of state losses and/or crime are found, BPK should immediately report the matter to the law enforcers in accordance with the legislation (Law No.15 of 2004) for them to follow up with an enquiry or investigation [1]. Investigative audits are considered reactive because they are conducted after the discovery of an initial indication of an irregularity. An investigative audit can originate from the result of a financial audit, performance audit, or audit with specific objectives [1,2]. BPK may conduct an investigative audit upon BPK's own initiative or at the request of authorised institutions such as the Corruption Eradication Commission, Police, Judiciary, and House of People's Representatives.

Over the years, BPK's investigations have revealed the increasing number of fraud cases in state finances. Based on BPK's Summary Report of Semester II/2017, until 31 December 2017, BPK had issued 16 investigative audit reports, with indications of state/regional losses amounting to IDR 5.18 trillion, as shown in Figure 1.

The Deputy Chairman of the BPK for the period 2017–2019, Mr Barullah Akbar, stated that the BPK has taken several corrective steps to strengthen the impact of the audit results in order to address public doubts on the quality of the BPK's audits. This was done in an effort to address public concerns. BPK will enhance quality assurance in order to improve

the influence that the audit results have on the organisation, which will be accomplished through improving audit quality. Ms Saskia Stuiveling, President of the Netherlands Court of Audit, has also expressed a similar view when he asked BPK to increase its investigative audit capabilities in order to reduce instances of corruption involving state finances [3].

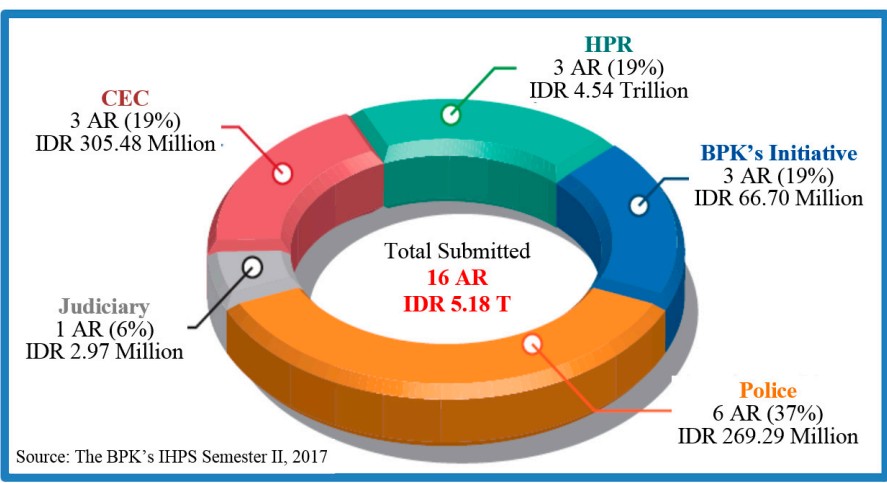

**Figure 1.** Investigative Audit Results as of 31 December 2017 [4].

Following this concern, this study aims to examine the factors influencing investigative audit quality in BPK. Specifically, two factors are chosen, namely, auditor competency and digital forensic support. This study is important, as the current literature has limited evidence on the factors that can influence investigative audit quality. This study assists in closing one of the gaps in the literature by examining auditor competency and digital forensic support in a specific context of audits, that is, investigative audit quality. In addition, the findings of this study can assist BPK in improving audit quality so that it can increase the impact of the audit results through the enhancement of quality assurance. The next section provides the literature review. This is followed by Section 3, which presents the research design, and Section 4, which provides the results. The final section, Section 5, concludes this study.

## 2. Literature Review

### 2.1. Investigative Audit Quality

Audit quality, in general, is a combination of the detection and disclosure of fraud on financial statements [2,3,5]. These studies suggested that audit output influences audit quality because audit quality is based on stakeholders' views. Value-added or quality auditing is emerging as one of the most powerful tools for continuous quality improvement [4]. Value-added audit services are defined as client-service activities resulting from an audit that are not directly related to verifying the financial statements, providing important benefits for both clients and audit firms [6]. From the meaning of value-added auditing, as stated by the researchers, it can be concluded that a quality audit is an audit that can improve the quality of clients continuously. One of the indicators of a quality audit is an audit that is able to reveal the indications loss of the state and/or criminal elements in order to realise the creation of good and transparent governance.

The conformity of audit planning procedures with audit objectives, an acceptable limit of audit failure, and the auditor's ability to detect fraud and report it in audit reports are all components of investigative audit quality [2,7–10]. It is made up of components that have the potential to influence the quality of the outcomes, such as the auditor's level of professionalism and independence, the auditor's personal credibility, and the sufficiency of the audit evidence to support audit reports [11–13]. Therefore, the evaluation of the quality of the investigative audit was based on a number of criteria and indicators, including the professionalism of the investigative auditors, the process of implementation,

and the reporting. This method of measuring auditor professionalism requires the auditor to have knowledge and capabilities in the field of investigation, to have experience in conducting investigative audits, and to be independent from all activities that are reviewed. Infrastructure that supports investigative audits, adheres to established standards, and has an understanding of entity audit risks are all required components of the implementation process of investigative audits. In terms of reporting, it entails reporting the results of investigative audits, such as revealing the presence of fraud, reporting the shortcomings of internal supervision, and drawing conclusions regarding the alleged criminal activity.

Several investigations into the quality of audit investigations have been conducted, and those studies have concentrated on the factors that influence audit quality [14–16]. For example, one study has investigated audit quality attributes in the UK by polling the finance directors of 210 companies that were listed in the UK through the use of a questionnaire survey consisting of 29 auditor characteristics. The study shows that five primary determinants of audit quality emerged, namely, firm reputation, the level of technical competence possessed by the audit partners, the level of integrity possessed by the firm, and the quality of the working relationship with the audit partner [14]. In another study, the study found 56 items relating to audit firm factors which include reputation, capability, responsiveness, expertise, and experience [17].

In earlier studies on audit quality, considerations pertaining to service quality were also incorporated [15,18]. As a direct result of this, a model for audit quality that is known as the AUDITQUAL model was developed [14]. The quality of the audit is broken down into two distinct parts, namely, the technical qualities and the service qualities, according to this model [14]. According to one study, the technical qualities include competence and independence, whereas the service qualities include responsiveness, non-audit services, and understanding. Both sets of qualities are important [18]. On the other hand, the quality of the investigations was not the primary focus of these studies. It is possible that a better understanding of the factors that examine investigative audit performance can be gained by looking at such a study through the lens of a different context, such as investigative audit performance. As a result, auditor competency and digital forensic support were chosen as the two factors to focus on in this study. These two factors being chosen is consistent with the current scenario in Indonesia that highlighted the concern regarding the investigative auditors' competency due to the political conflicts and whether digital forensic support can improve the investigative audit quality in BPK.

*2.2. Auditor Competency*

Studies have suggested that one of the factors that affected the overall quality of the investigative audit is auditor competency. Competency is defined as "a state of expertise sufficient to achieve explicit audit objectives" [19] (p. 1171). The ability to demonstrate the knowledge, expertise, and skills of each individual in order to achieve the audit objectives in a continuous manner relates to auditor competency [19–23]. According to previous studies, auditor competency can be measured by the individual's capacity to understand the audit entity's processes and capacities, as well as by having specialised expertise and developed knowledge [23–27]. The following factors and indicators were used to determine an auditor's level of competency: (1) knowledge of business process entities (including entity operational processes, entity management processes, and entity support processes); (2) special skills (including investigative auditor certification and digital forensic expertise); and (3) ability (being able to learn problems quickly).

The audit literature has suggested the need to have auditor competency in maintaining audit quality [28,29]. In order for auditors to be able to support audit performance, they need to be competent [29]. This competency can be obtained and improved upon thanks to two factors, namely, experience and education [30]. Auditor competency is a qualification that is required of auditors in order for auditors to successfully carry out audits [31,32]. It is necessary to engage in ongoing professional education in order to achieve these competencies. Personal qualities, general knowledge, and specialised abilities are the three

aspects that contribute to an auditor's overall competency. In addition, in order to generate attitudes that are creative and full of innovation, an auditor needs to possess a number of other qualities in addition to intelligence [33–35]. These qualities include a high level of commitment, good behaviour, and a good imagination. These studies generally shared the assumption that level of ability in detecting fraud is a factor in maintaining the quality of investigative audits

Studies that have examined auditor competency have used various variables to measure auditor competency [23–27]. The variables can be divided into three dimensions, namely, knowledge of business process entities, special skills, and ability. For the first dimension, knowledge of business process entities, three types of processes are determined. The three processes are entity operational processes, entity management processes, and entity support processes. For the second dimension, special skills, studies have identified two fields of expertise: investigative auditor certification and digital forensic skills. The last dimension is an ability which includes self-learning and the ability to solve problems efficiently.

A good audit is the execution of a well-designed audit process by motivated and trained auditors who understand the inherent uncertainty of the audit and appropriately adjust to the specific conditions of the client [8]. In other words, a good audit is the execution of a well-designed audit process by motivated and trained auditors. In other words, an audit is considered good if it satisfies the criteria for what constitutes a "good" audit. In addition, increasing the auditor's competency results in auditors having deeper knowledge and giving better judgement, both of which are required to accomplish the goal of achieving a high level of audit quality [36]. On the basis of the findings of the researchers who came before them, one is able to reach the following conclusion: competency on the part of auditors has a positive impact on the quality of investigative audits. As a consequence of this, the following hypothesis is developed:

**Hypothesis 1.** *Auditor competency significantly and positively influences investigative audit quality.*

### 2.3. Digital Forensic Support

Digital forensics is defined as "the use of scientifically derived and proven methods toward the preservation, collection, validation, identification, analysis, interpretation, documentation, and presentation of digital evidence derived from digital sources for the purpose of facilitation or furthering the reconstruction of events found to be criminal, or helping to anticipate unauthorised actions shown to be disruptive to planned operations" ([34], p. 2). Digital forensic support refers to a methodology that protects, collects, validates, identifies, analyses, interprets, documents, and presents digital evidence that originates from digital sources with the intention of reconstructing criminal offences that can be used as evidence in court. This methodology was developed in order to aid in the investigation and prosecution of criminal cases [36,37]. According to the findings of previous research, the term "digital forensic support" refers to activities that involve the acquisition, testing, analysis, and presentation of electronic evidence that is stored digitally on digital equipment such as computers, audio players, cellular phones, facsimile machines, and others [36,38–40]. This includes: (1) the acquisition of digital evidence (digital evidence search; digital evidence recognition; digital evidence collection and documentation); (2) the testing of digital evidence (real digital evidence; digital data filtering; digital data validation); and (3) the analysis and presentation of digital evidence (analysing hidden data; determining the significance of the digital data obtained; reconstructing the digital data obtained). It is possible to measure it using these methods.

The disclosure level of the government of Indonesia's financial statements is still relatively low. This is despite the fact that financial statements have been recognised as a tool of quality in the government's financial management [40]. As a result of this fact, a forensic audit is required in order to determine whether or not the governmental financial statements are, in fact, transparent and accountable [41]. There is an ever-increasing demand for the assistance of digital forensic experts in the fight against fraud because of

the growing prevalence of the use of digital tools to conceal fraudulent activity in today's society. This is one of the main reasons why there is such a demand. According to one study, digital forensic is an application in the field of computer science and technology that is used to acquire legal proof [42]. In order to obtain digital evidence that can be used to apprehend the criminals responsible for the offence, its goal is to provide scientific evidence of high-tech computer crime. This evidence will be gathered in order to bring those responsible for the offence to justice. In addition, digital forensic support directly has an effect that is beneficial on the level of fraud detection, which was discovered in another body of the audit literature [43].

Studies on audit quality have identified various variables to measure digital forensic support [1,18,38,39]. These variables can be divided into three dimensions. The first dimension is digital evidence acquisition. Under this dimension, three types of digital evidence are identified, namely, digital evidence search, digital evidence acknowledgment, and the collection and documentation of digital evidence. The second dimension is digital evidence testing, which includes real digital proof, digital data filtering, and digital data validation. The last dimension is digital data analysis and presentation. In this dimension, it includes analysing hidden data, determining the significance of the obtained digital data, and reconstructing the obtained digital data.

The development of information and communication technologies also has the potential to improve the quality of audits, according to a number of studies that have been conducted on the subject. For instance, information and communication systems are becoming an increasingly fertile ground for the production of electronic evidence, also known as e-evidence, which can be utilised in audits, investigations, or litigation [44]. This type of evidence can be utilised in any of these situations. Because the role of the auditor in the detection of fraud has grown over the years, it is now essential to apply the same principle and conduct the audit using the computer in order to collect digital data [44]. This is because the role of the auditor has grown over the years. In addition, any case involving fraud should at the very least consider the possibility of using cyber or digital evidence and the potential value that this evidence could bring to the case [45–49]. This should be done regardless of whether or not the evidence in question is actually used. Arguably, it is possible to draw the following conclusion about the application of digital forensics: it has a positive effect on the quality of the investigative audit. As a consequence of this, the following hypothesis is developed:

**Hypothesis 2.** *Digital forensic support significantly and positively influences investigative audit quality.*

### 3. Research Design
#### 3.1. Respondents

For the purpose of this study, the investigative auditors in the BPK were chosen to participate as respondents. These individuals were chosen because they have prior experience carrying out investigative audits, either with or without having certification as a Certified Fraud Auditor/Certified Forensic Auditor (CFrA/CFA). In addition, their selection was based on the fact that they have a CFA or CFRA. It has been determined that there are a total of 150 heads of representative auditors working for BPK. Due to the relatively small population, the researchers decided to approach all the investigative auditors as respondents in this study.

#### 3.2. Research Instrument

This study was conducted with the aid of a questionnaire survey as the instrument of enquiry. The development of the questionnaire was based on reviewing past studies, with some modifications to suit the context of this study. The questionnaire is divided into four sections. In the first section, Section B, the respondents are asked to fill out their demographic profile by providing information such as their age, gender, and position. In the

second section of the survey, the respondents were given a series of questions concerning auditor competency. In this section, the respondents were requested to identify their knowledge of business process entities, special skills, and ability. There are 14 questions related to auditor competency.

In Section C, the respondents are asked to complete a series of questions pertaining to digital forensic support. In this section, the respondents were requested to identify their digital evidence acquisition, testing of digital evidence, and analysis of the presentation of digital evidence. There are 18 questions related to digital forensic support. In the last section of the questionnaire, Section D, the respondents were requested to complete a series of questions pertaining to investigative audit quality. The respondents responded to questions regarding audit professionalism, the implementation process of investigation audits, and reporting the results of investigation audits. There are 12 questions related to investigative audit quality. In total, there are 44 questions in the questionnaire survey related to the independent variables and dependent variable (see Appendix A: Questionnaire). The respondents were requested to complete sections two through four of the questionnaire using a 5-point Likert scale. The variables in the questionnaires were measured based on Table 1.

**Table 1.** Variable Measurements.

| Variable | Dimension | Indicator | Scale |
|---|---|---|---|
| **Independent Variable** | | | |
| Auditor competency [22,24–26] | Knowledge of business process entities | Entity operational process | Semantic |
| | | Entity managerial process | |
| | | Entity support process | |
| | Special skills | Have an investigative auditor certification | |
| | | Have digital forensic skills | |
| | Ability | Self-update ability | |
| | | Efficient problem solving | |
| Digital forensic support [33,37,50] | Digital evidence acquisition | Digital evidence search | Semantic |
| | | Digital evidence acknowledgment | |
| | | Collection and documentation of digital evidence | |
| | Testing of digital evidence | Real digital proof | |
| | | Digital data filteration | |
| | | Digital data validation | |
| | Analysis of presentation of digital evidence | Analyse hidden data | |
| | | Determining the significance of digital data | |
| | | Reconstructing acquired digital data | |
| **Dependent Variable** | | | |
| Investigative audit quality [10–12] | Audit professionalism | Knowledge and ability in the field of investigation | Semantic |
| | | Experience in conducting investigative audits | |
| | Implementation process of investigation audit | Infrstructure supporting the conduct of investigative audits | |
| | | Follow standards | |
| | Reporting results of invesitgative audit | Disclosing fraud and suspected criminal acts | |
| | | Reporting internal control weaknesses | |

### 3.3. Data Collection

The questionnaires were distributed to the respondents via direct visits and emails to the respondents. Since the researchers can easily obtain access to the respondents, the researchers managed to get all the heads of the representative auditors to participate in the questionnaire survey. The data collection was completed within 2 months; no reminders were sent to the respondents to complete their questionnaire early. Hence, the researchers believed that there is no significant difference between those respondents who completed the questionnaire earlier and those who submitted later. In total, 150 questionnaires were completed and returned. Specifically, 60 respondents were from the representative offices and 90 respondents were from the head offices.

### 3.4. Data Analyses

In this study, quantitative methods and probability statistics were used to analyse sample data. The findings of this study applied to the population by testing the significance level using t-statistics based on a confidence interval of 95% and a risk of error of less than 5%.

This study began by classifying the response scores provided by the respondents into a number of different categories. This was ordered according to the maximum and minimum possible scores, as shown in Table 2. The average score of the variables was calculated with the help of descriptive statistics, which were applied through the construction of a frequency distribution table.

**Table 2.** Categorisation of Scores.

| Average Index | Category |
| :---: | :---: |
| 1.00–1.80 | Very Bad |
| 1.81–2.60 | Bad |
| 2.61–3.40 | Fair |
| 3.41–4.20 | Good |
| 4.21–5.00 | Very Good |

The research hypotheses were examined through the use of the Structural Equation Modeling (SEM) technique and the statistical programme Lisrel. Confirmatory factor analysis was used to evaluate the validity and reliability of the instrument by dividing the construct or latent variables into dimensions and indicators. These dimensions and indicators were then tested individually. A variable is considered to be legitimate if its t-factor exceeds the critical value (t-value is less than 1.96) and its loading factor is less than 0.70 [46]. Having a loading factor that is less than 0.50 is very significant in order for the indicator to be considered valid. A composite reliability measure and a variance extracted measure were used to test the instrument's reliability. A value of Construct Reliability (CR) greater than 0.70 and a value of Variance Extracted (VE) less than 0.50 were determined to indicate good reliability for the instrument [47].

The flowchart of the research model that was used to investigate the influence of independent variables (exogenous) on dependent variables (endogenous) is depicted in Figure 2. The research model was formulated in a structural model as follows:

$$\eta_1 = \gamma_{11}\ \xi_1 + \gamma_{21}\ \xi_2 + \zeta \tag{1}$$

Description: $\xi_1$ = auditor competency variable; $\xi_2$ = digital forensic support variable; $\eta1$ = the quality of investigation audit variable; $\gamma$ = path coefficient between exogenous latent variables; and $\zeta$ = measurement error of endogenous latent variables.

The concept of SEM was used to develop the various stages of data analysis that were used in this study. Before testing the hypotheses, the researchers used the Maximum

Likelihood method to estimate the parameters of the research model. They then evaluated the Goodness of Fit between the data and the research model.

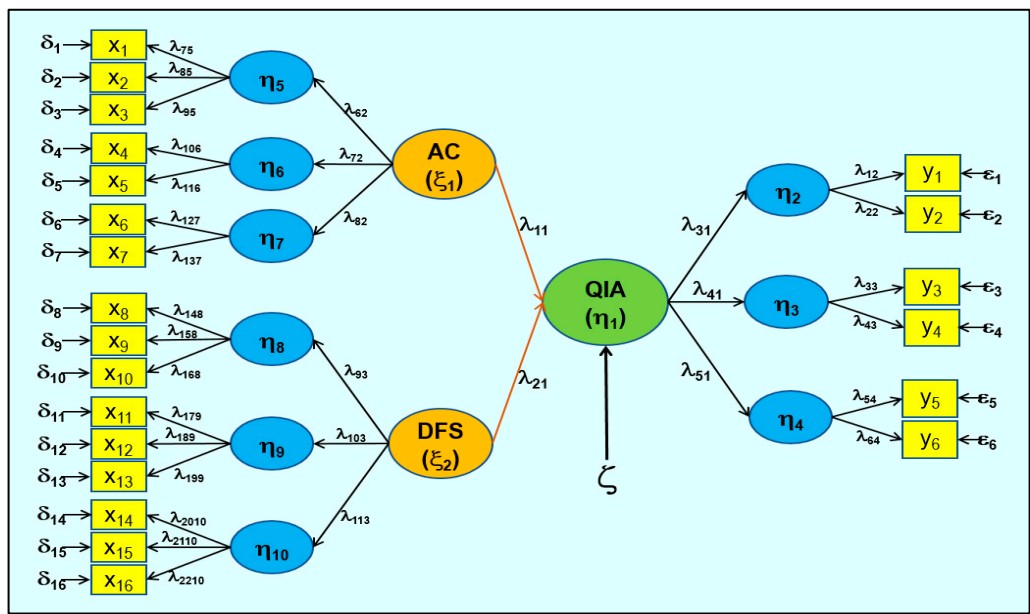

**Figure 2.** Flowchart Research Model.

## 4. Results

### 4.1. Descriptive Statistics

Investigative audit quality is the dependent variable, and the results of the descriptive statistics on the independent variables, auditor competency, and digital forensic support are presented in Table 3. There are three different statements regarding auditor competency. When it comes to auditor competency, the statement pertaining to special skills received the highest mean score from the respondents, which was 4.40. This was followed by the statement pertaining to ability, which received a mean score of 4.30, and the statement pertaining to knowledge of business process entities, which received a mean score of 4.21.

**Table 3.** Descriptive Statistics and Scores.

| No | | Variable | Σ Score | Mean | Categorisation |
|---|---|---|---|---|---|
| 1 | | Auditor Competency (AC) | 4503.50 | 4.29 | Very good |
| | a. | Knowledge of business process entities (KBPE) | 1895.50 | 4.21 | Very good |
| | b. | Special skills (SS) | 1319.00 | 4.40 | Very good |
| | c. | Ability (Abi) | 1289.00 | 4.30 | Very good |
| 2 | | Digital forensic support (DFS) | 5835.00 | 4.32 | Very good |
| | a. | Digital evidence acquisition (DEA) | 1950.00 | 4.17 | Good |
| | b. | Testing of digital evidence (TDE) | 1289.00 | 4.33 | Very good |
| | c. | Analysis and presentation of digital evidence (APDE) | 2006.00 | 4.46 | Very good |

**Table 3.** *Cont.*

| No | | Variable | Σ Score | Mean | Categorisation |
|---|---|---|---|---|---|
| **3** | | Investigative audit quality (QIA) | 3726.00 | 4.14 | Good |
| | a. | Audit professionalism (AP) | 1310.00 | 4.37 | Very good |
| | b. | Implementation process of investigative audit (IPIA) | 1225.00 | 4.08 | Good |
| | c. | Reporting results of investigative audit (RRIA) | 1191.50 | 3.97 | Good |

The analysis and presentation of digital evidence received the highest mean score for digital forensic support, with a mean score of 4.46. The testing of digital evidence received the next-highest mean score, with a mean score of 4.33, and digital evidence acquisition received the third-highest mean score, with a mean score of 4.17. If we look at auditor competency and digital forensic support side by side, we can see that the latter has a mean score that is 4.32 points higher. The mean score that corresponds to audit professionalism is the one that corresponds to the highest quality of the investigative audit (4.37), while the mean score that corresponds to reporting the results of the investigation audit is the one that corresponds to the lowest quality of the investigative audit (3.97). As can be seen in Table 2, the overall score of the variables, as well as their average rating, were given the ratings of "good and very good" based on the responses that were given.

*4.2. Confirmatory Factor Analysis (CFA)*

Figure 3 provides a visual representation of the CFA examination of auditor competency. This exogenous variable was evaluated based on the results of measurements taken across a total of three dimensions and seven distinct indicators. Figure 3 is a visual representation of the CFA model in its second order.

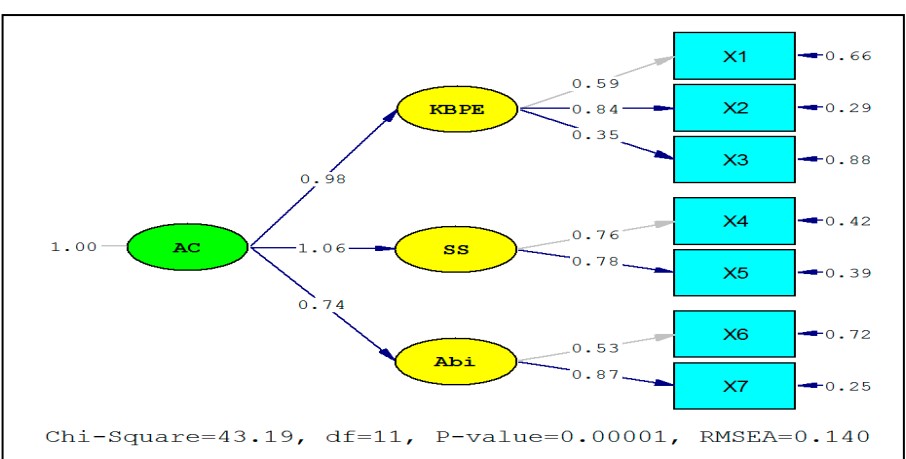

**Figure 3.** CFA Test of AC Variable.

Due to the fact that the loading factor of the X3 indicator was lower than 0.5, it needs to be eliminated from the model and re-specified in the manner shown in Figure 4. In a nutshell, all of the indicators gave precise readings of the AC Variable because the loading factors of each indicator were greater than 0.5.

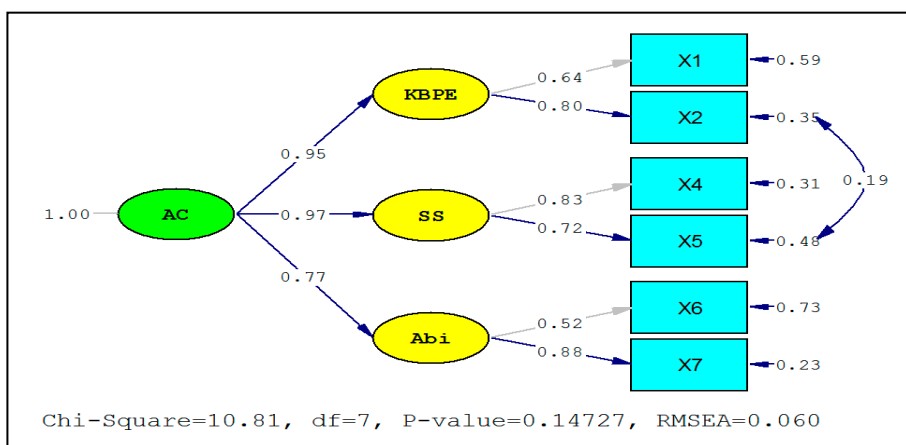

**Figure 4.** CFA Test of AC Re-specification.

Table 4 displays the outcomes of the first-order test conducted on the KBPE, SS, and Abi dimensions. Based on the results of the measurements taken across each dimension, it was determined that all of the indicators were valid, reliable, and consistent. This was demonstrated by loading factor values that were greater than 0.5, CR values that were close to 0.7, and VE value values that were greater than 0.5. The second-order test on AC for each dimension produced measurements of AC that were valid, reliable, and consistent. These measurements had a loading factor that was greater than 0.5, a CR that was greater than 0.7, and a VE that was greater than 0.5. Abi had the lowest loading factor, which made it the least effective at reflecting the AC Variable. In comparison, the SS Dimension had the highest loading factor, which made it the most effective at reflecting the AC Variable.

**Table 4.** Validity and Reliability Test Results of AC Re-specification.

| Latent Variable | Indicator | $\lambda$ | $\lambda^2$ | $\varepsilon$ | CR | VE | Information |
|---|---|---|---|---|---|---|---|
| | | | First-Order | | | | |
| KBPE | X1 | 0.64 | 0.41 | 0.59 | 0.69 | 0.52 | Reliable |
| | X2 | 0.80 | 0.64 | 0.36 | | | |
| SS | X4 | 0.83 | 0.69 | 0.31 | 0.75 | 0.60 | Reliable |
| | X5 | 0.72 | 0.52 | 0.48 | | | |
| Abi | X6 | 0.52 | 0.27 | 0.73 | 0.67 | 0.52 | Reliable |
| | X7 | 0.88 | 0.77 | 0.23 | | | |
| | | | Second-Order | | | | |
| AC | KBPE | 0.95 | 0.90 | 0.10 | 0.93 | 0.81 | Reliable |
| | SS | 0.97 | 0.94 | 0.06 | | | |
| | Abi | 0.77 | 0.59 | 0.41 | | | |

Figure 5, which focuses on digital forensic support, displays the results of the CFA of decision forensic support. The CFA looked at digital forensic support. When analysing this exogenous variable, a total of three dimensions and nine distinct indicators were used. Figure 5 is a visual representation of the CFA model in its second order. Due to the fact that the loading factor of the TDE dimension was higher than 1, the model needed to be re-specified, as shown in Figure 6. In a nutshell, the loading factors that were greater than 0.5 demonstrated that each indicator was a valid measurement of the DSF variable. This was evidenced by the fact that the indicator served as a valid measurement of the DSF variable.

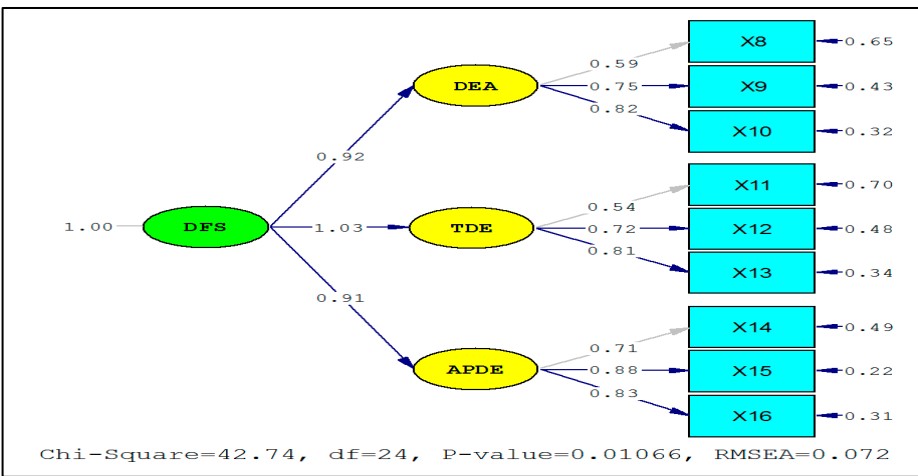

**Figure 5.** CFA Test of DFS Variable.

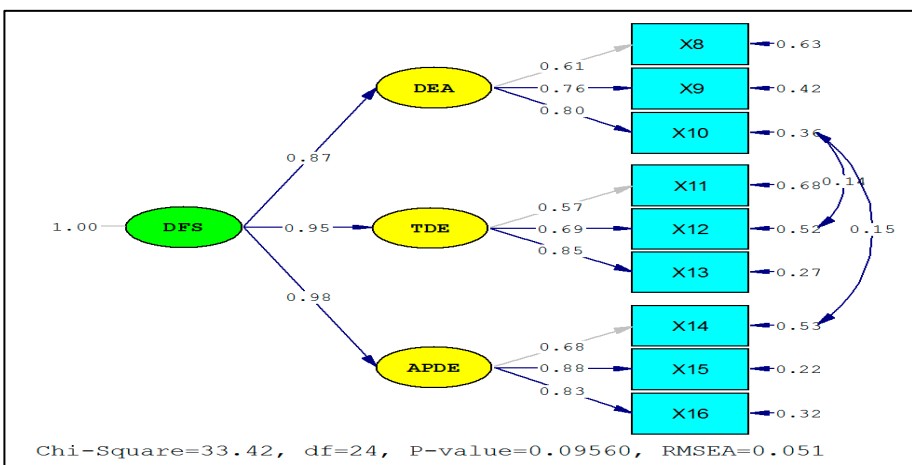

**Figure 6.** CFA Test of DFS Re-specification.

Table 5 contains the findings of the first-order test conducted on the DEA, TDE, and APDE dimensions. All of the indicators were shown to be valid, reliable, and consistent, with loading factor values greater than 0.5, CR value values greater than 0.7, and VE value values greater than 0.5, as shown by the measurements of each dimension. The second-order test on DFS discovered that the measurements of the DSF Variable were valid, reliable, and consistent across all dimensions, with a loading factor greater than 0.5, a CR greater than 0.7, and a VE greater than 0.5. This was determined by the test's conclusion that VE was greater than 0.5. The APDE Dimension had the highest loading factor, and as a consequence, it was the one that provided the most accurate reflection of the DFS Variable. The DEA, on the other hand, had the lowest loading factor, which led to it having the lowest accuracy. The endogenous variable of investigative audit quality was evaluated based on its performance across three dimensions and six indicators.

Figure 7 provides a visual representation of the CFA second-order model for the QIA Variable. As shown in Figure 8, the model needed to be re-specified in order to take into account the fact that the loading factor of AP was greater than 1. In sum, all of the indicators provided accurate measurements of the QIA Variable on account of the fact that their respective loading factors were greater than 0.5.

**Table 5.** Validity and Reliability Test Results of DSF Re-specification.

| Latent Variable | Indicator | $\lambda$ | $\lambda^2$ | $\varepsilon$ | CR | VE | Information |
|---|---|---|---|---|---|---|---|
| | | | First-Order | | | | |
| DEA | X8 | 0.61 | 0.37 | 0.63 | | | |
| | X9 | 0.76 | 0.58 | 0.42 | 0.77 | 0.53 | Reliable |
| | X10 | 0.80 | 0.64 | 0.36 | | | |
| TDE | X11 | 0.57 | 0.32 | 0.68 | | | |
| | X12 | 0.69 | 0.48 | 0.52 | 0.75 | 0.51 | Reliable |
| | X13 | 0.85 | 0.72 | 0.28 | | | |
| APDE | X14 | 0.68 | 0.46 | 0.54 | | | |
| | X15 | 0.88 | 0.77 | 0.23 | 0.84 | 0.64 | Reliable |
| | X16 | 0.83 | 0.69 | 0.31 | | | |
| | | | Second-Order | | | | |
| DFS | DEA | 0.87 | 0.76 | 0.24 | | | |
| | TDE | 0.95 | 0.90 | 0.10 | 0.95 | 0.87 | Reliable |
| | APDE | 0.98 | 0.96 | 0.04 | | | |

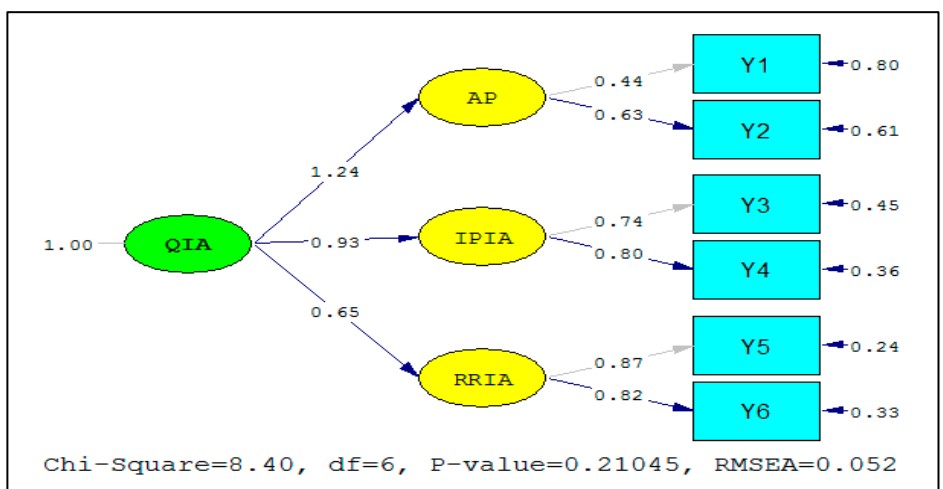

**Figure 7.** CFA Test of QIA Variable.

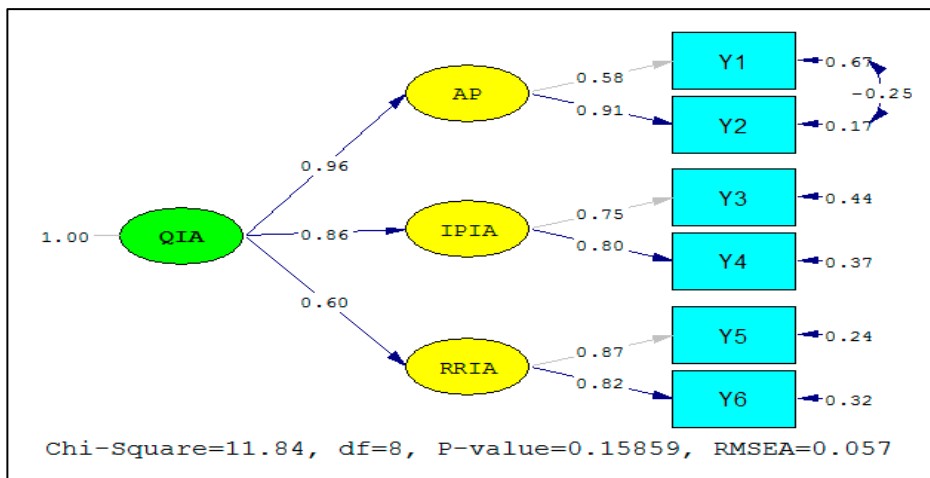

**Figure 8.** CFA Test of QIA Re-specification.

Table 6 contains the findings of the first-order test performed on the AP, IPIA, and RRIA dimensions. With loading factors greater than 0.5, CR values greater than 0.7, and VE values greater than 0.5, every one of the indicators met the criteria for validity, reliability, and consistency.

**Table 6.** Validity and Reliability Test Results of QIA Re-specification.

| Latent Variable | Indicator | $\lambda$ | $\lambda^2$ | $\varepsilon$ | CR | VE | Information |
|---|---|---|---|---|---|---|---|
| | | | First-Order | | | | |
| AP | Y1 | 0.58 | 0.34 | 0.66 | 0.73 | 0.58 | Reliable |
| | Y2 | 0.91 | 0.83 | 0.17 | | | |
| IPIA | Y3 | 0.75 | 0.56 | 0/44 | 0.75 | 0.60 | Reliable |
| | Y4 | 0.80 | 0.64 | 0.36 | | | |
| RRIA | Y5 | 0.87 | 0.76 | 0.24 | 0.83 | 0.71 | Reliable |
| | Y6 | 0.82 | 0.67 | 0.33 | | | |
| | | | Second-Order | | | | |
| QIA | AP | 0.96 | 0.92 | 0.08 | 0.86 | 0.67 | Reliable |
| | IPIA | 0.86 | 0.74 | 0.26 | | | |
| | RRIA | 0.60 | 0.36 | 0.64 | | | |

The second-order test on QIA discovered that the measurements of the QIA Variable were valid, reliable, and consistent across all dimensions, with a loading factor greater than 0.5, a CR greater than 0.7, and a VE greater than 0.5. This was determined by the fact that all three of these values were greater than 0.5. RRIA had the lowest loading factor, which resulted in it having the least accurate reflection of the QIA Variable. In contrast, the AP Dimension had the highest loading factor, which resulted in it having the most accurate reflection of the QIA Variable.

### 4.3. Full Structural Model

The SEM was used to make estimates for the evaluation of the fitted model and the parameter values. In this research, the empirical model that was generated from the theoretical model needs to be subjected to comprehensive model testing. Following the CFA for each latent variable, the full SEM was carried out, as shown in Figure 9.

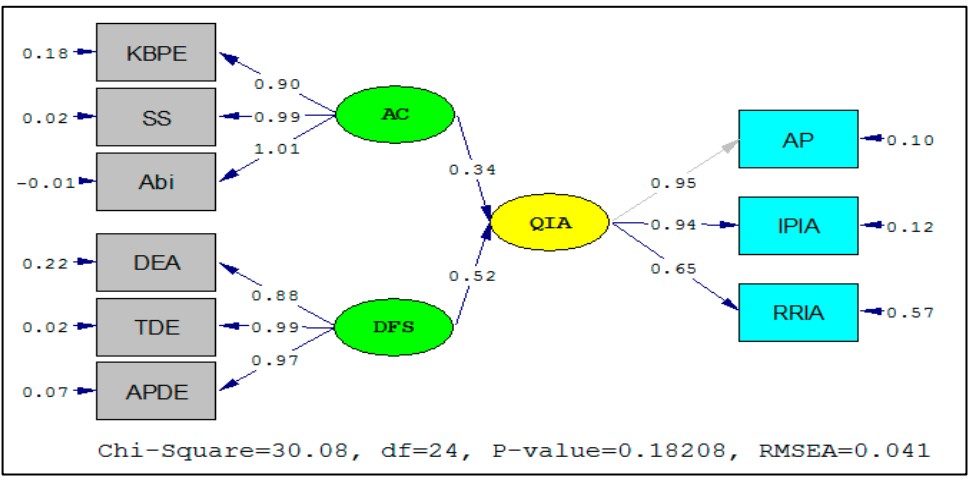

**Figure 9.** Full Structural Model.

Figure 9 reveals that there is still an indicator—specifically, the Abi Dimension—that possesses a value of factor loading that is greater than 1, and this indicator can be found

in the table. As a result of this, the specifications of the Full Structural Model needed to be revised, as shown in Figure 10. In addition to this, the results of the Lisrel that are based on the re-specified Full Structural Model produce the following mathematical structural equation:

$$QIA = 0.35AC + 0.51DFS + 0.37 \tag{2}$$

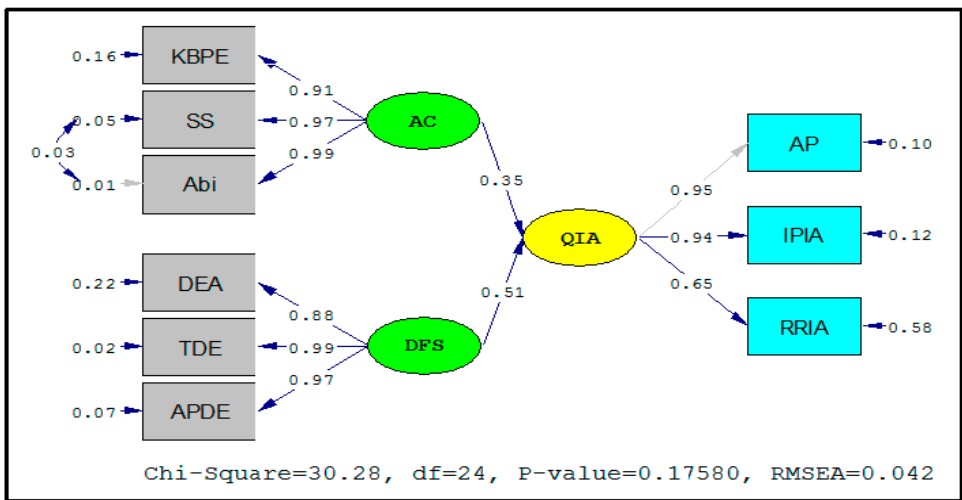

**Figure 10.** Re-specifications of the Full Structural Model.

In order to determine whether the model was appropriate and fair, a comprehensive evaluation of the model was carried out, making use of the conformity and hypothesis models. Table 7 contains the findings, which are organised according to the Goodness of Fit indices presented there. The outcome of the suitability testing performed on the overall model revealed that all Goodness of Fit indices had satisfied the prerequisites for the fit. This was the case after the testing had been performed. As the Lisrel result of the full structural model re-specification, which can be seen displayed in Figure 11, the Path Coefficient summary of the relationship between the variables is presented in Table 8.

**Table 7.** Evaluation of Fit of Full Structural Model Re-specification.

| No | Goodness of Fit | Target Value | Value | Description |
|----|----------------|--------------|-------|-------------|
| 1 | Chi-square (*p* value) | Expected small (≥0.05) | 30.28 (0.17580) | Small (Fit) |
| 2 | RMSEA | ≤0.08 | 0.042 | Fit |
| 3 | NFI | ≥0.90 | 0.99 | Fit |
| 4 | NNFI | ≥0.90 | 0.99 | Fit |
| 5 | CFI | ≥0.90 | 1.00 | Fit |
| 6 | IFI | ≥0.90 | 1.00 | Fit |
| 7 | RFI | ≥0.90 | 0.98 | Fit |
| 8 | SRMR | ≤0.05 | 0.021 | Fit |
| 9 | GFI | ≥0.90 | 0.96 | Fit |
| 10 | AGFI | ≥0.90 | 0.92 | Fit |

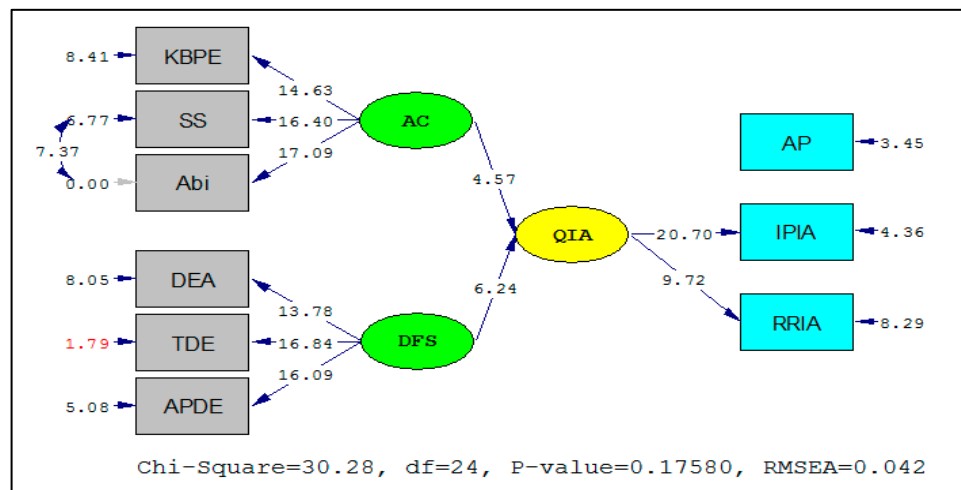

**Figure 11.** Re-specifications of the Full Structural Model (T-value).

**Table 8.** Results of Path Coefficient Estimates and Statistical Tests.

| Relationship | Path Coefficient | T-Value | R-Square (Simultan) |
|---|---|---|---|
| AC ➡ QIA | 0.35 | 4.57 | 0.63 |
| DFS ➡ QIA | 0.51 | 6.24 | |

Sixty-three percent of the total effect that the variables AC and DFS had on the QIA Variable was accounted for by that percentage. The remaining 37% was determined by other factors, which did not include the variables that were evaluated based on their degree of independence. According to the Path Coefficient, the variable that had the most significant influence on QIA was DFS, which had a path value of 0.51, followed by AC, which had a path value of 0.35. Both of these variables were considered to be independent variables.

*4.4. Hypothesis Testing*

The hypothesis *t*-test statistics provided that H0 is rejected if $t_{value} > 1.96$ or $-t_{value} < -1.96$ for $\alpha = 0.05$ in the 95% confidence interval. In relation to the first hypothesis, the result of hypothesis testing is as follows:

| | |
|---|---|
| $H_0: \gamma_{11} = 0$ | Auditor competency does not significantly influence investigative audit quality. |
| $H_1: \gamma_{11} \neq 0$ | Auditor competency significantly influences investigative audit quality. |
| Lisrel Result | $t_{value} = 4.57$; hence, $H_0$ Rejected and $H_1$ Accepted |

In line with the findings in earlier studies, this result offered empirical evidence demonstrating that auditor competency significantly and positively influences investigative audit quality [7,26,30].

In relation to hypothesis 2, the result of hypothesis 2 is as follows:

| | |
|---|---|
| $H_0: \gamma_{21} = 0$ | Decision forensic support does not significantly influence investigative audit quality |
| $H_1: \gamma_{21} \neq 0$ | Decision forensic support significantly influences investigative audit quality |
| Lisrel Result | $t_{value} = 6.24$; hence, $H_0$ Rejected and $H_1$ Accepted |

In line with the findings of previous studies, this result provided empirical evidence that the use of digital forensic support had a significant positive influence on the investigative audit quality [43–45].

*4.5. Discussion*

The results of the path coefficient significance test on the structural model show the hypothesis testing that auditor competency significantly and positively influences the investigative audit quality. This is evidenced by the t-count value exceeding the threshold of 1.96 at the 95% significance level. Auditor competency in this study consists of three dimensions, namely: entity business process knowledge, special skills, and capability. Based on the ranking of each standard factor load value, the dimension that most reflects auditor competency is capability, followed by special skills and entity business process knowledge. The findings of this study show an improved capability through increasing the ability of the auditors to study and resolve problems efficiently and effectively. The findings also suggest that special skills are necessary to improve investigative audit quality through increasing expertise and obtaining auditor certification. In terms of entity business process knowledge, this can be carried out through increasing the auditor's understanding of the management and operational processes of the audited entity. The findings of this study are similar to findings in previous studies that showed that auditor competence is one of the factors that affect the level of fraud detection [27,28,35,48,51].

The results of the path coefficient significance test on the structural model in this study also show the hypothesis testing that digital forensic support significantly and positively influences investigative audit quality. This is evidenced by the t-count value exceeding the threshold of 1.96 at the 95% significance level. Digital forensic support has three dimensions, namely, digital evidence acquisition, digital evidence testing, and digital evidence analysis and presentation. Based on the ranking of each standard factor load value, the dimension that most reflects digital forensic support is digital evidence testing, followed by digital evidence testing, digital evidence analysis and presentation, and, finally, digital evidence acquisition.

The findings of this study show that digital evidence testing can influence investigative audit quality by increasing the certainty that the digital data obtained are real evidence and by performing filtering and validation of the digital data. In terms of digital evidence analysis and presentation, it is reflected by increasing the activities of analysing hidden data, determining the significance of the obtained digital data, and reconstructing the obtained digital data. With regard to digital testing evidence, it is reflected by increasing the search for digital evidence, obtaining entity recognition of the digital evidence, and collecting and documenting the digital evidence. The findings in this study show that the three dimensions of digital forensic support are relevant to investigative audit quality. The findings of this study are consistent with previous studies that explained that digital forensic support is one of the factors that can influence investigative audit quality [1,18,37,38].

**5. Conclusions**

The purpose of this study is to investigate whether auditor competency and digital forensic support influence investigative audit quality. Based on the findings shown in this study, auditor competency and digital forensic support are both significant factors that have the potential to influence the quality of investigative audits performed in BPK. Therefore, it is the responsibility of BPK and any other parties connected with this transaction to ensure that their auditors have sufficient auditing expertise. The findings in this study suggest that the auditors need to improve their knowledge and understanding of business processes regarding the entity to be audited, improving quality and capability. This can be done by obtaining periodic certification of expertise in order to increase their knowledge and capabilities. Another suggestion for improving auditor competency is to cooperate with the audit team by determining audit targets such as agreeing on audit deadlines, determining the allocation and workload of team members by involving all team elements in decision making so that they can design flexible audit procedures, always reviewing and revising audit targets in accordance with conditions that occur in the field, and carrying out strong and inherent supervision in order to meet the expectations of the tasks.

This study also found that digital forensic support significantly and positively influences investigative audit quality. In other words, in ensuring that the investigative auditors can be able to conduct high-quality investigations and audits, it is essential for them to have access to a reliable digital forensic support tool in their respective offices.

This study is important, as its findings can provide insight to investigative auditors to enhance their knowledge and understanding of the audited entity business processes, improving quality and capability through the periodic certification of expertise. This would allow them to carry out their responsibilities effectively and detect fraudulent financial reporting through financial audits. Ultimately, this overall effort is expected to improve the quality of investigative audits.

This study is not without limitations. Firstly, the scope of this investigation is limited to just two categories: auditor competency and digital forensic support. As a result, future studies may include additional variables and other factors such as workload pressure and time intensity, which may have the potential to influence the findings of investigative audit quality. Secondly, this study focuses only on the investigative auditors in BPK. Hence, it is recommended to re-examine according to the findings of this study, using the same research method but in different units of analysis, so that the generalisability of the findings in this study can be improved. Thirdly, this study used a questionnaire survey to achieve the objectives of this study, and, hence, the findings in this study represent the subjective measures of auditor competency and digital forensic support. Perhaps future studies can use objective measures such as experiments so that the findings can be confirmed using other research designs.

In sum, the findings of this study provide understanding of the factors influencing investigative audit quality. This is a fundamental thing in the activities of auditing institutions, such as the BPK, as an institution that has the authority to conduct audits of the management and accountability of state finances. The findings also serve as a fundamental understanding of sustaining work integrity and professionalism. The findings of this study can be used as a discourse by other audit institutions of a similar nature in order to enhance the quality of the findings produced by their investigative audit activities.

**Author Contributions:** Conceptualization, S.M.; Methodology, C.S.; Writing—original draft, H.S.; Writing—review & editing, E.K.G. All authors have read and agreed to the published version of the manuscript.

**Funding:** This research received no external funding. However, the APC was funded by Universitas Padjadjaran, Bandung, Indonesia.

**Informed Consent Statement:** Informed consent was obtained from all subjects involved in the study.

**Conflicts of Interest:** The authors declare no conflict of interest.

### Appendix A. Questionnaire

**INVITATION TO PARTICIPATE IN THE STUDY OF THE INFLUENCE OF AUDITOR COMPETENCY AND DIGITAL FORENSIC SUPPORT ON INVESTIGATIVE AUDIT QUALITY**

| | |
|---|---|
| Principal Researcher: | Hendra Susanto |
| Researcher's Contact: | Faculty of Economics and Business |
| | Universitas Padjadjaran |
| | Bandung, Indonesia |
| Email: | hendrasusanto1972@yahoo.co.uk |

We are a group of researchers from Universitas Padjadjaran, Indonesia and Universiti Teknologi MARA, Malaysia who are currently embarking on a project entitled "The Influence of Auditor Competency and Digital Forensic Support on Investigative Audit Quality".

This study aims to examine the influence of auditor competency and digital forensic support on investigative audit quality in the Audit Board of the Republic of Indonesia (BPK). The findings of this study can assist BPK in selecting and identifying appropriate auditors to carry out their duties, which will increase the effectiveness and efficiency of BPK. The success of this study is highly dependent on your response to our survey. We therefore request your participation in completing this survey questionnaire.

For information, the information provided will only be used for academic purposes. This study report will not identify any person by name unless permission is given. Data analysis will be conducted and reported in such a way that the information cannot be linked directly to any person in the public sector.

## SECTION A

Demographic Profile

| | |
|---|---|
| Name | : … … … … … … … … … … … |
| Age | : … … … … … … … … … … … … years |
| Gender | : ☐ Male ☐ Female |
| Department | : … … … … … … … … … … … |
| Education | : … … … … … … … … … … … |
| Faculty/Discipline | : … … … … … … … … … … … |
| Professional Certification | : ☐ Yes ☐ No |
| Investigative Audit Experience | : ☐ 1–3 years ☐ 4–7 years ☐ > 7 years |

## SECTION B

This section requests that the respondents provide their response on auditor competency. Please complete the series of questions by circling your response on the 5-point Likert scale below.

| No. | Item | Response |
|---|---|---|
| 1 | Investigative auditors first have to understand the operational processes of the audited entity. | 1 2 3 4 5 — Extremely disagree ← → Extremely agree |
| 2 | The investigative auditor has previously sought information on laws and regulations related to the entity's operations. | 1 2 3 4 5 — Extremely disagree ← → Extremely agree |
| 3 | Investigative auditors should endeavour to study the management policies of the audited entity. | 1 2 3 4 5 — Extremely disagree ← → Extremely agree |
| 4 | Investigative auditors need to try to profile the internal and external environment and stakeholders of the entity to be audited. | 1 2 3 4 5 — Extremely disagree ← → Extremely agree |

| No. | Item | Response |
|---|---|---|
| 5 | Understanding the entity's support processes is an important thing for an investigative auditor to do. | 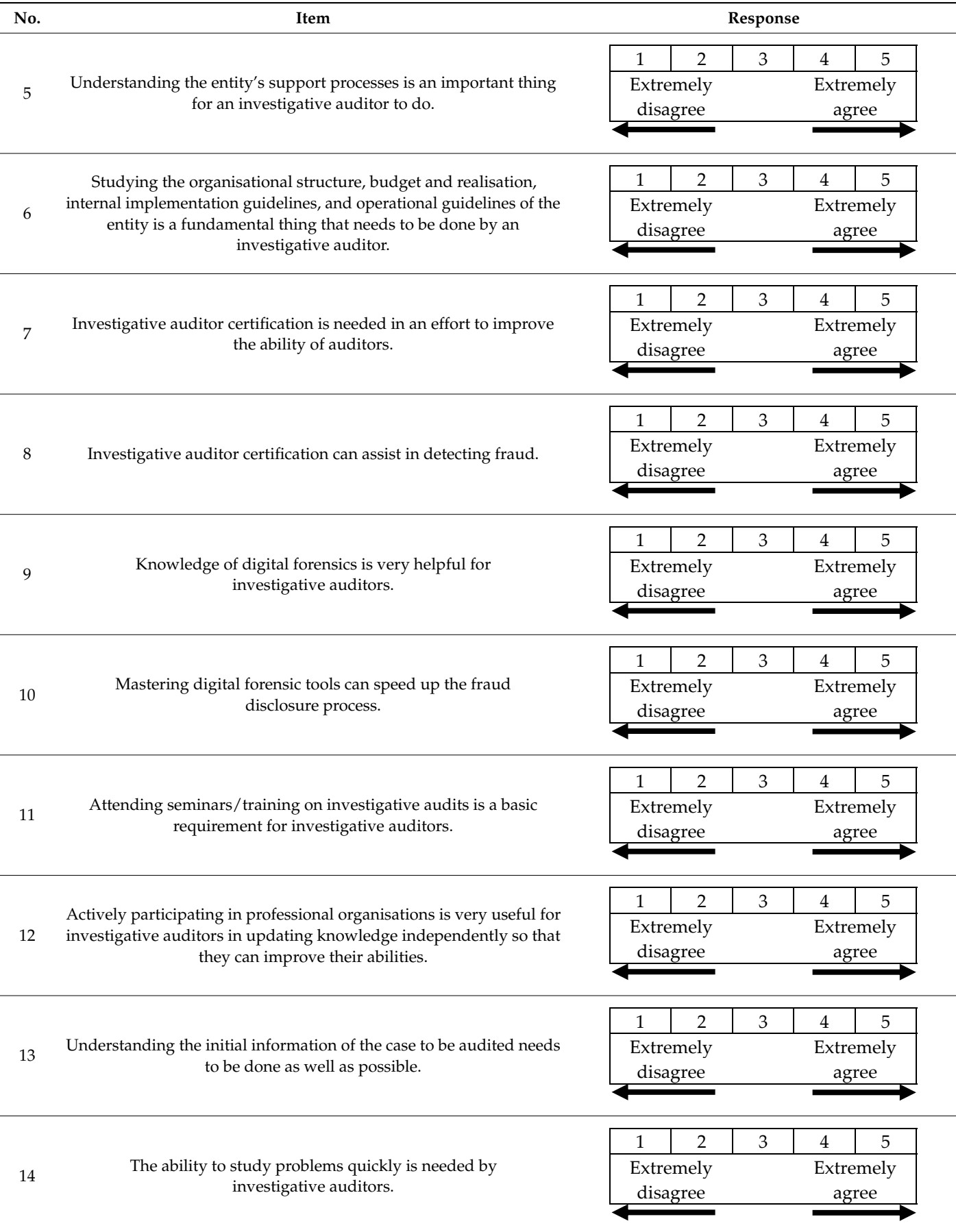 |
| 6 | Studying the organisational structure, budget and realisation, internal implementation guidelines, and operational guidelines of the entity is a fundamental thing that needs to be done by an investigative auditor. | |
| 7 | Investigative auditor certification is needed in an effort to improve the ability of auditors. | |
| 8 | Investigative auditor certification can assist in detecting fraud. | |
| 9 | Knowledge of digital forensics is very helpful for investigative auditors. | |
| 10 | Mastering digital forensic tools can speed up the fraud disclosure process. | |
| 11 | Attending seminars/training on investigative audits is a basic requirement for investigative auditors. | |
| 12 | Actively participating in professional organisations is very useful for investigative auditors in updating knowledge independently so that they can improve their abilities. | |
| 13 | Understanding the initial information of the case to be audited needs to be done as well as possible. | |
| 14 | The ability to study problems quickly is needed by investigative auditors. | |

SECTION C

This section requests that that respondents provide their response on digital forensic support. Please complete the series of questions by circling your response on the 5-point Likert scale below.

| No. | Item | Response | | | | |
|---|---|---|---|---|---|---|
| 1 | The use of digital forensic equipment support will make it easier for investigative auditors to find audit evidence. | 1 | 2 | 3 | 4 | 5 |
| | | Extremely disagree | | | Extremely agree | |
| 2 | The support of digital forensic equipment will save time in searching for digital evidence. | 1 | 2 | 3 | 4 | 5 |
| | | Extremely disagree | | | Extremely agree | |
| 3 | The acquisition of digital evidence must be able to guarantee its authenticity, integrity, and availability. | 1 | 2 | 3 | 4 | 5 |
| | | Extremely disagree | | | Extremely agree | |
| 4 | In order for digital evidence to be recognised, it must be carried out by a digital forensic expert. | 1 | 2 | 3 | 4 | 5 |
| | | Extremely disagree | | | Extremely agree | |
| 5 | The digital evidence obtained must be carefully recorded and easily recognisable. | 1 | 2 | 3 | 4 | 5 |
| | | Extremely disagree | | | Extremely agree | |
| 6 | The digital evidence obtained must be documented in a secure place. | 1 | 2 | 3 | 4 | 5 |
| | | Extremely disagree | | | Extremely agree | |
| 7 | Digital evidence must be tested for reliability and can be accounted for to support information in an audit process. | 1 | 2 | 3 | 4 | 5 |
| | | Extremely disagree | | | Extremely agree | |
| 8 | Digital evidence must be accessible, displayed, and guarantee integrity. | 1 | 2 | 3 | 4 | 5 |
| | | Extremely disagree | | | Extremely agree | |
| 9 | In conducting the acquisition of digital evidence, not all evidence obtained must be collected and stored so that it can support the audit process. | 1 | 2 | 3 | 4 | 5 |
| | | Extremely disagree | | | Extremely agree | |

| No. | Item | Response |
|-----|------|----------|
| 10 | The digital evidence obtained must be filtered according to certain fraud detections only. | 1 / 2 / 3 / 4 / 5 — Extremely disagree ← → Extremely agree |
| 11 | Investigative auditors must validate digital data obtained from digital forensic experts. | 1 / 2 / 3 / 4 / 5 — Extremely disagree ← → Extremely agree |
| 12 | Always validate digital evidence to test its reliability. | 1 / 2 / 3 / 4 / 5 — Extremely disagree ← → Extremely agree |
| 13 | With the support of digital forensics, investigative auditors can obtain evidence hidden by the perpetrators of fraud. | 1 / 2 / 3 / 4 / 5 — Extremely disagree ← → Extremely agree |
| 14 | Analysis of digital evidence must be carried out by a digital forensic expert. | 1 / 2 / 3 / 4 / 5 — Extremely disagree ← → Extremely agree |
| 15 | The significance of the digital data obtained can affect the level of fraud detection. | 1 / 2 / 3 / 4 / 5 — Extremely disagree ← → Extremely agree |
| 16 | The significance of digital data is determined by the analysis and presentation of digital evidence. | 1 / 2 / 3 / 4 / 5 — Extremely disagree ← → Extremely agree |
| 17 | Reconstruction of the digital data obtained will support the analysis and presentation of digital evidence. | 1 / 2 / 3 / 4 / 5 — Extremely disagree ← → Extremely agree |
| 18 | By reconstructing the digital data obtained, it will make it easier for investigative auditors to analyse and present digital evidence. | 1 / 2 / 3 / 4 / 5 — Extremely disagree ← → Extremely agree |

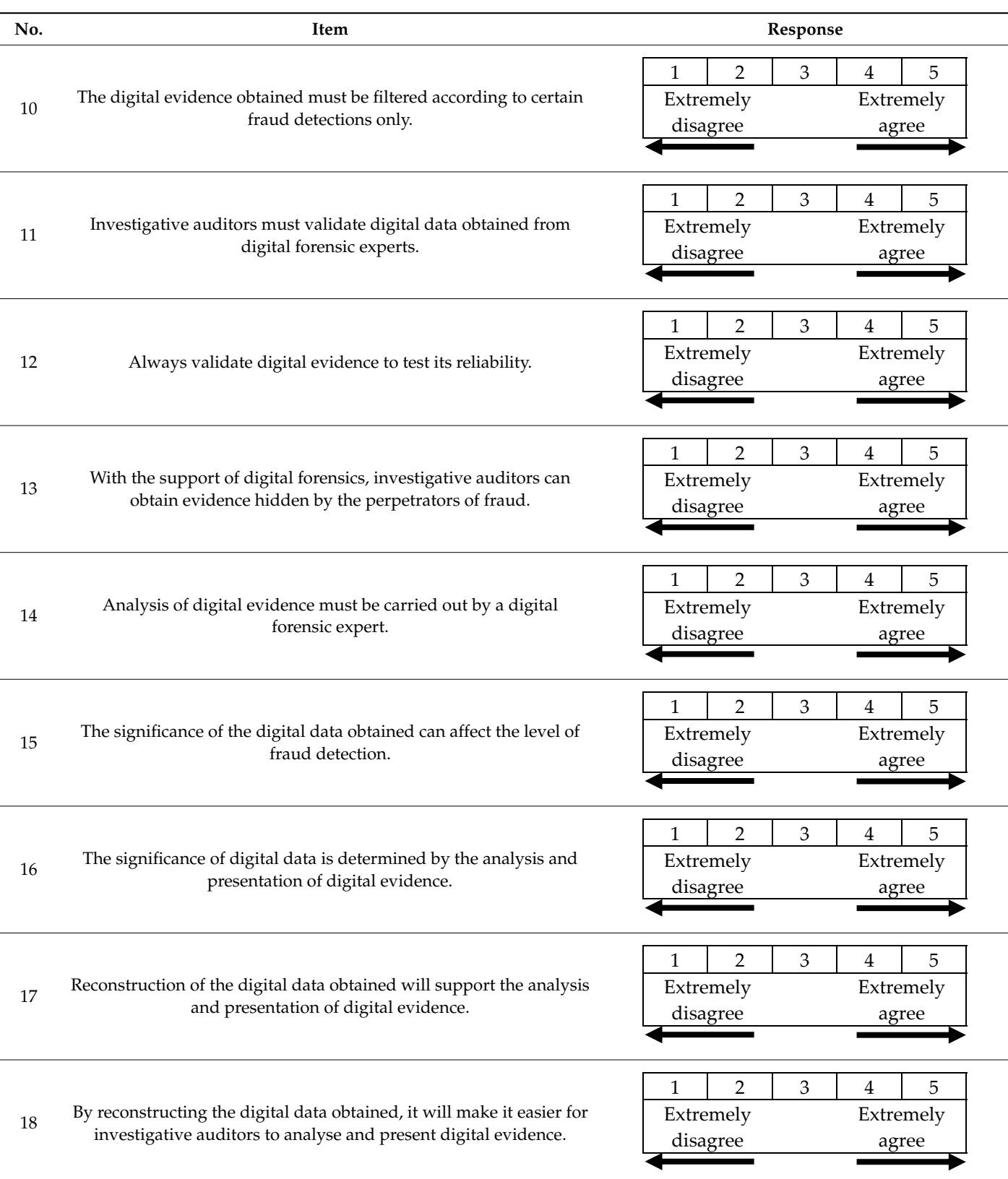

SECTION D

This section requests that the respondents provide their response on investigative audit quality. Please complete the series of questions by circling your response on the 5-point Likert scale below.

| No. | Item | Response |
|-----|------|----------|
| 1 | Auditors who have knowledge and abilities in the investigative field are more professional in detecting fraud that will affect the quality of investigative audit results. | 1 2 3 4 5 — Extremely disagree ← → Extremely agree |
| 2 | Knowledge and ability in the investigative field can improve the quality of investigative audit results. | 1 2 3 4 5 — Extremely disagree ← → Extremely agree |
| 3 | Experience in conducting investigative audits can increase the professionalism of investigative auditors in detecting fraud. | 1 2 3 4 5 — Extremely disagree ← → Extremely agree |
| 4 | The quality of an investigative audit is highly dependent on the experience of the auditor in detecting fraud. | 1 2 3 4 5 — Extremely disagree ← → Extremely agree |
| 5 | Complete facilities and infrastructure can assist in detecting fraud and improving the quality of investigative audit results. | 1 2 3 4 5 — Extremely disagree ← → Extremely agree |
| 6 | In detecting fraud carried out without being supported by complete facilities and infrastructure, it is still possible to carry out investigative audits with satisfactory results. | 1 2 3 4 5 — Extremely disagree ← → Extremely agree |
| 7 | In detecting fraud, it is possible not to follow audit standards as long as the quality of audit results is met. | 1 2 3 4 5 — Extremely disagree ← → Extremely agree |
| 8 | In carrying out an investigative audit, always follow the audit standards that have been set to produce a quality audit. | 1 2 3 4 5 — Extremely disagree ← → Extremely agree |
| 9 | Disclosure of fraud detection in investigative audit reports will affect the quality of investigative audit results. | 1 2 3 4 5 — Extremely disagree ← → Extremely agree |
| 10 | The report on the results of the investigative audit should be able to conclude that there is a suspected criminal act committed by the management of the entity. | 1 2 3 4 5 — Extremely disagree ← → Extremely agree |

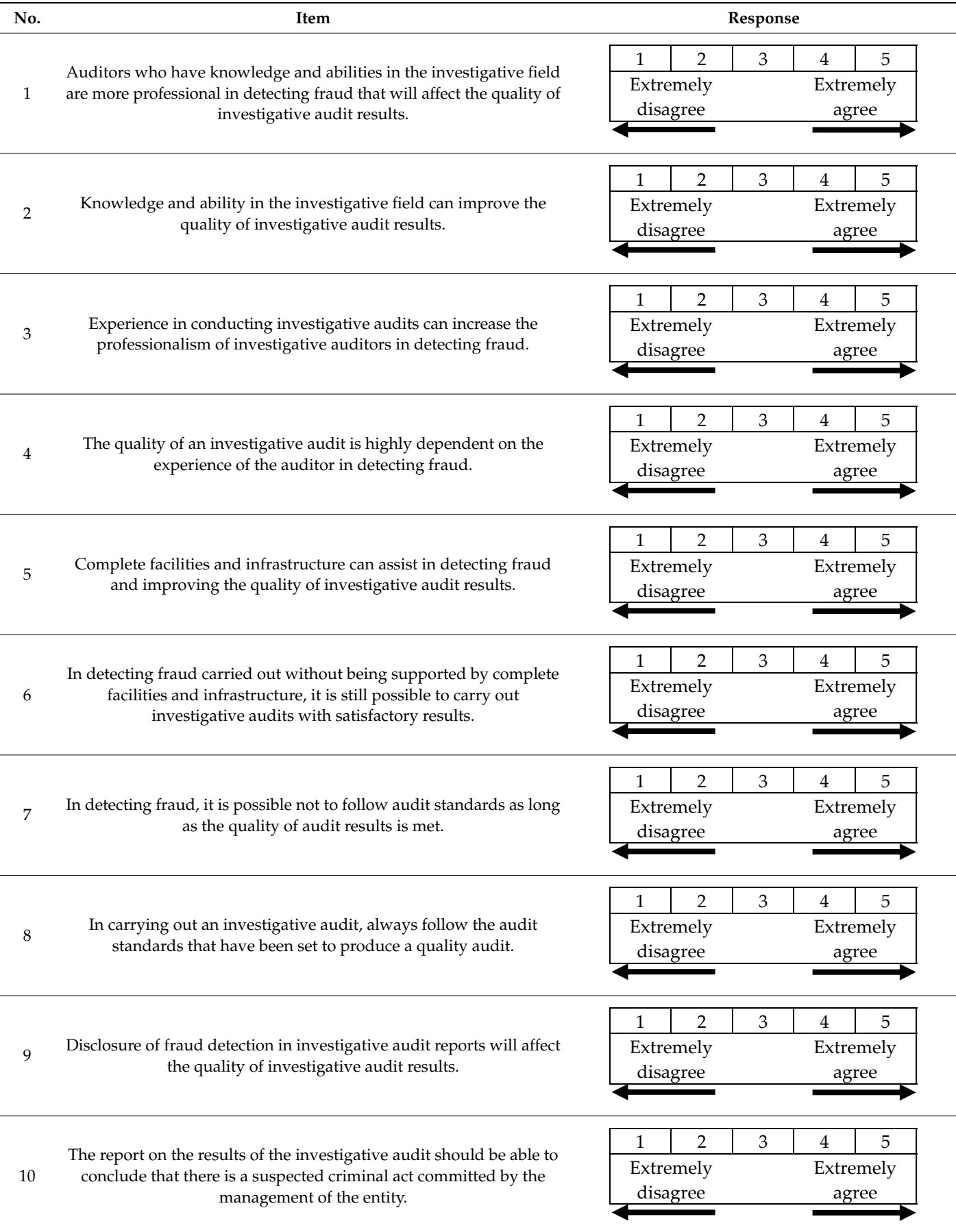

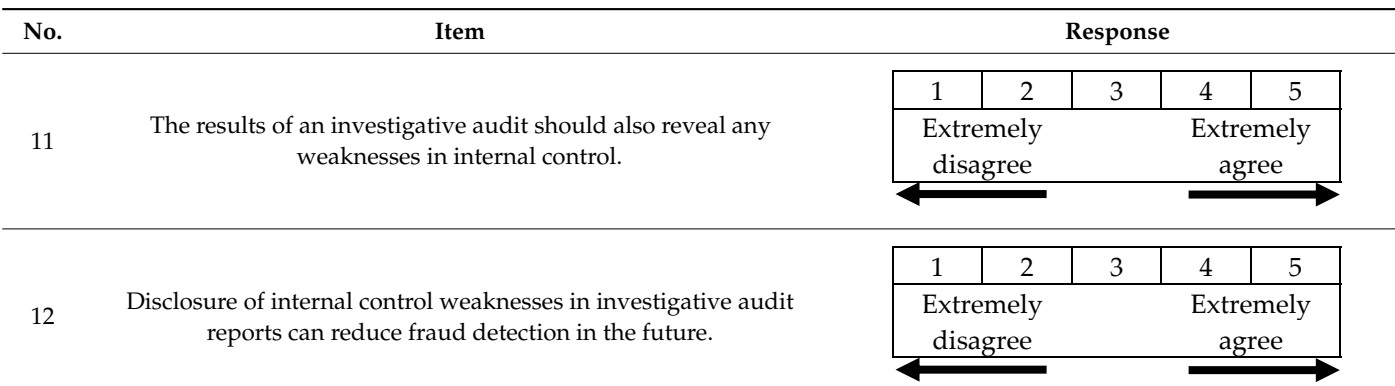

| No. | Item | Response | | | | |
|---|---|---|---|---|---|---|
| 11 | The results of an investigative audit should also reveal any weaknesses in internal control. | 1 | 2 | 3 | 4 | 5 |
| | | Extremely disagree | | | Extremely agree | |
| 12 | Disclosure of internal control weaknesses in investigative audit reports can reduce fraud detection in the future. | 1 | 2 | 3 | 4 | 5 |
| | | Extremely disagree | | | Extremely agree | |

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
