# Peer review of "Sustaining Investigative Audit Quality through Auditor Competency and Digital Forensic Support: A Consensus Study"

_sustainability, doi:10.3390/su142215141_

Round 1

Reviewer 1 Report

This study, we investigate the factors that influence the quality of investigative audits. This study shows that both factors significantly 17 and positively influence investigative audit quality.

In my opinion the paper is well structure, but the originality and the value of the paper is not obvious.  In addition, I have some concerns about the methodology that was applied. For example, it does not mention how the variables were measured. This results in a major weakness of this study. Moreover, the author(s) have to present in more detail the selection of the sample. Were there any differences between early and late responders?

Furthermore, the author(s) have to present better the motivation of the study, to discuss the results with previous studies, and to present more analytically the limitations and directions for future research. The concussions’ section is very weak, as well as the literature review. The figures are not necessarily appeared in the text but perhaps in an appendix.

I hope my comments have been helpful.

Author Response

Dear Reviewer

I have revised the manuscript based on your comments. We appreciate your review on the revised manuscript.

Thank you

Reviewer 2 Report

1. It is not clear whether the three different statements regarding auditor competence, Knowledge of business process entities, Special skills and Ability, are established by the authors or are taken from scholarly literature. The same observation applies to the Digital forensic support and Investigative audit quality.

 2. It would have been interesting to compare the results obtained in this research with those in other studies from different countries or periods.

Author Response

Dear Reviewer

We have revised the manuscript based on your comments. Attached is the response to reviewer form.

Thank you

Reviewer 3 Report

The paper investigates the influence of auditor competency and digital forensic support on the quality of investigative audits. The authors find that both factors have a positive effect on the investigative audits' quality.

I think the paper deals with an important and up-to-date topic. Although this study contains significant and interesting material, I recommend the following changes:

1. In the Introduction, the gap needs to be improved.  The authors mention the importance of investigative audits. However, it is important to mention how this study differs from the existing ones and how this paper contributes to the existing literature. 

2. The choice of the two factors (auditor competency and digital forensic support) must be explained. Why were these two factors chosen?

3. The authors should also consider whether the hypotheses must be rewritten. The reasoning of the hypotheses indicates a positive relationship between auditor competency and digital forensic support and audit quality. However, the authors do not predict any sign (positive or negative) for H1 and H2. So, we have two possible approaches: first, rewrite the hypotheses so as to predict the sign; second, explain why no sign can be predicted for H1 and H2.

4. The literature review (section 2) should be updated. The most recent paper cited is from 2017...

5. The methodology needs also to be more grounded. The authors must explain how the questionnaire was constructed. Was the questionnaire based on previous literature?

6. The discussion of the results should be improved. The authors present the results but there is almost no discussion about them. 

Finally, there are also some spelling and standardization errors (see, for example, lines 29-30 on the first page) and some lack of citations (see, for example, lines 107 and 108 on page 3).

Author Response

Dear Reviewer

We have revised the manuscript based on your comments. Please see the attached form.

Thank you

Round 2

Reviewer 1 Report

Thank you for the revised paper. 

Authors should present their measurements (items, scales) in detail. Table 1 is not enough. A copy of the questionnaire could be presented in the appendix.

Author Response

Dear Reviewer

I have further revised the paper according to your comments. Thank you

Reviewer 3 Report

I think the manuscript has been sufficiently improved to warrant publication in Sustainability.

Author Response

Dear Reviewer 

I have revised the paper according to your comments. Thank you

Round 3

Reviewer 1 Report

Thanks for your revised paper.

Good luck to your paper!